# LEARNING TO IMPUTE: A GENERAL FRAMEWORK FOR SEMI-SUPERVISED LEARNING

## ABSTRACT

Recent semi-supervised learning methods have shown to achieve comparable results to their supervised counterparts while using only a small portion of labels in image classification tasks thanks to their regularization strategies. In this paper, we take a more direct approach for semi-supervised learning and propose learning to impute the labels of unlabeled samples such that a network achieves better generalization when it is trained on these labels. We pose the problem in a learning-to-learn formulation which can easily be incorporated to the state-of-the-art semi-supervised techniques and boost their performance especially when the labels are limited. We demonstrate that our method is applicable to both classification and regression problems including image classification and facial landmark detection tasks.

## 1 INTRODUCTION

Semi-supervised learning (SSL) (Chapelle et al., 2009) is one of the approaches to learn not only from labeled samples but also unlabeled ones. Under certain assumptions such as presence of smooth prediction functions that map data to labels, of low-dimensional manifolds that the high-dimensional data lies (Chapelle et al., 2009), SSL methods provide a way to leverage the information at unlabeled data and lessens the dependency on labels. Recent work (Tarvainen & Valpola, 2017; Miyato et al., 2018; Berthelot et al., 2019; Xie et al., 2019) have shown that semi-supervised learning by using only a small portion of labels can achieve competitive results with the supervised counterparts in image classification tasks (*i.e.* CIFAR10, SVHN). They built on a variation of the well-known iterative bootstrapping method (Yarowsky, 1995) where in each iteration a classifier is trained on the current set of labeled data, the learned classifier is used to generate label for unlabeled data. However, the generalization performance of this approach is known to suffer from fitting the model on wrongly labeled samples and overfitting into self-generated labels (Tarvainen & Valpola, 2017). Thus, they mitigate these issues by various regularization strategies.

While there exist several regularization (Srivastava et al., 2014) and augmentation (Zhang et al., 2018; DeVries & Taylor, 2017) techniques in image recognition problems which are known to increase the generalization performance of deep networks, specific regularization strategies for semi-supervised classification are used to estimate correct labels for the unlabeled data by either encouraging the models to produce confident outputs (Lee, 2013; Grandvalet & Bengio, 2005) and/or consistent output distributions when its inputs are perturbed (Tarvainen & Valpola, 2017; Miyato et al., 2018; Berthelot et al., 2019). The assumption here is that if a good regularization strategy exists, it can enable the network to recover the correct labels for the unlabeled data and then the method can obtain similar performance with the supervised counterpart when trained on them. Though this ad-hoc paradigm is shown to be effective, it raises a natural question for a more *direct* approach: Can we instead encourage the network to label the unlabeled data such that the network achieves better generalization performance when trained with them?

In this paper, we propose a new learning-to-learn method for semi-supervised learning that can be put in a meta-learning framework to address this question. Our method involves learning an update rule to label unlabeled training samples such that training our model using these predicted labels improves its performance not only on itself but also on a meta-validation set. Crucially, our method is highly generic and can easily be incorporated to the state-of-the-art methods (Lee, 2013; Berthelot et al., 2019) and boost their performance, in particular, in the regime where the number of available labels is limited. Moreover, our method is not limited to classification problems, we show that it can

be extended to semi-supervised regression tasks where the output space is continuous and achieves significant performance gains.

## 2 RELATED WORK

**Semi-supervised classification.** There is a rich body of literature in SSL (Chapelle et al., 2009) for classification. Most of the recent work (Lee, 2013; Tarvainen & Valpola, 2017; Miyato et al., 2018; Berthelot et al., 2019; Xie et al., 2019) builds on the idea of the bootstrapping technique of Yarowsky (1995) that involves iterative optimization of classifier on a set of labeled data and refinement of its labels for the unlabeled data. This paradigm is known to overfit on the noisy self-generated labels and thus suffer from low generalization performance. To alleviate the sensitivity to inaccurate labeling, researchers introduce various regularization strategies. Grandvalet & Bengio (2005) propose a minimum entropy regularizer that encourages each unlabeled sample to be assigned to only one of the classes with high probability. Lee (2013) instead follow a more direct approach, use the predicted label with the maximum probability for each unlabeled sample as true-label which is called "pseudo-label". An orthogonal regularization strategy is to encourage a classifier to be invariant to various stochastic factors and produce consistent predictions for unlabeled data when the noise is added to intermediate representations (Srivastava et al., 2014) and to input in an adversarial manner (Miyato et al., 2018). In the same vein, Laine & Aila (2017); Tarvainen & Valpola (2017) introduce the Π-model and Mean-Teacher that are regularized to be consistent over previous training iterations by using a temporal ensembling and teacher/student networks respectively. Recently Berthelot et al. (2019) introduced MixMatch algorithm that further extended the previous work by unifying the idea of the consistency regularization and augmentation strategies (Zhang et al., 2018). While the recent techniques are shown to be effective in several classification benchmarks, the idea of consistency regularization is still implicit in terms of generalization performance. Here we take an orthogonal and more direct approach to this approach and learn to impute the labels of the unlabeled samples that improve the generalization performance of a classifier. We also show that our method can be used with the recent SSL techniques.

**Semi-supervised regression.** Some of the recent techniques in semi-supervised classification are shown to be applicable to regression problems. Jean et al. (2018) have adapted various existing SSL methods such as label propagation, VAT (Miyato et al., 2018) and Mean Teacher (Tarvainen & Valpola, 2017) and studied their performance in regression. The same authors also proposed a Bayesian SSL approach for SSL regression problems which is based on the recent deep kernel learning method (Wilson et al., 2016) based approach for semi-supervised learning that aims at minimizing predictive variance of unlabeled data. As in the classification task, these methods are typically ad-hoc and does not aim to generate labels for the unlabeled data that are optimized to improve the generalization in regression.

**Meta-learning.** Our method is also related to meta-learning (Schmidhuber, 1987; Bengio et al., 1992) and inspired from the recent work (Andrychowicz et al., 2016; Finn et al., 2017) where the goal is typically to quickly learn a new task from a small amount of new data. Andrychowicz et al. (2016); Finn et al. (2017) propose a learn gradient through gradient approach to train a model which has a strong generalization ability to unseen test data and can be easily adapted to new/unseen tasks. Sung et al. (2018) introduce the relation network to learn an additional metric such that the learned model's feature embedding can be generalized for unseen data and unseen tasks. Ren et al. (2018b) adopt the meta learning to learn the weight of samples to tackle the sample imbalance problem. Lorraine & Duvenaud (2018); MacKay et al. (2019) employ meta learning for automatically learning the hyper-parameters of the deep neural networks. Meta-learning has recently been applied to unsupervised learning (Hsu et al., 2019) and SSL for few shot learning (Ren et al., 2018a). Ren et al. (2018a) adapts the prototypical network to use unlabeled examples when producing prototypes, enabling the prototypical network to exploit those unlabeled data to assist few shot classification. Antoniou & Storkey (2019) propose to learn a label-free loss function, parameterized as a neural network that enables the classifier to leverage the information from a validation set and achieves better performance in few-shot learning. (Li et al., 2019) propose a meta-learning technique to initialize a self-training model and to filter out noisy pseudo labels for semi-supervised few-shot learning. Similarly, our work also builds on the ideas of optimizing for improving the generalization for unseen samples. However, in contrast to the existing meta-learning methods that is proposed for few-shot

learning problems, we focus on semi-supervised learning in general classification and regression problems where the number of samples are not limited to few.

## 3 METHOD

Consider a dataset $\mathcal{D}$ that consists of $|\mathcal{L}|$ labeled samples $\mathcal{L} = \{(\boldsymbol{x}_1, \boldsymbol{y}_1), (\boldsymbol{x}_2, \boldsymbol{y}_2), \dots, (\boldsymbol{x}_{|\mathcal{L}|}, \boldsymbol{y}_{|\mathcal{L}|})\}$ which is further split into a training $\mathcal{T}$ and meta-validation set $\mathcal{V}$ with $|\mathcal{T}|$ and $|\mathcal{V}|$ samples respectively. We also have a set of unlabeled samples $\mathcal{U} = \{\boldsymbol{x}^u\}_{i=1,\dots,|\mathcal{U}|}$. Further, we let $\Phi_\theta$ denote a model (a function parameterized for instance, as a deep neural network) with parameters $\theta$, that is trained to predict labels $y$ from given samples $x$ as $\boldsymbol{y} = \Phi_\theta(\boldsymbol{x})$.

We are interested in imputing missing labels $\boldsymbol{z}$ of the unlabeled samples that are not only accurate but also improve performance at inference time, when included in our training set for optimizing model parameters $\theta$. A straightforward approach to this problem is to first train a model to optimize the following cost function on the training set $\mathcal{T}$:

$$\arg\min_\theta \sum_{\boldsymbol{x}\in\mathcal{T}} \ell(\Phi_\theta(\boldsymbol{x}), \boldsymbol{y}) \tag{1}$$

where $\ell$ is a task specific loss, such as a soft-max followed by a cross-entropy loss for classification or squared loss for regression tasks.

One can then use the trained model $\Phi_\theta$ to impute labels for samples in the unlabeled set $\mathcal{U}$ as $\boldsymbol{z} = \Phi_\theta(\tilde{\boldsymbol{x}}^u)$ as in Tarvainen & Valpola (2017); Berthelot et al. (2019), where $\tilde{\boldsymbol{x}}^u$ denotes randomly perturbed input image $\boldsymbol{x}^u$ (*e.g.* random crops). Note that the label imputation procedure can be replaced with other pseudo-label prediction techniques.

Now we can expand $\mathcal{T}$ by adding the unlabeled samples with their imputed pseudo-labels *i.e.* $\mathcal{A} = \mathcal{T} \cup \mathcal{U}$ and further train $\theta$ by using Eq. (1). However, as many pseudo-labels $\boldsymbol{z}$ will likely be noisy, there is no guarantee that training on the augmented train set will improve the performance of the model on the meta-validation set $\mathcal{V}$:

$$C^V = \sum_{\boldsymbol{x}\in\mathcal{V}} \ell(\Phi_\theta(\boldsymbol{x}), \boldsymbol{y}). \tag{2}$$

Note that in all our experiments $\mathcal{V}$ is randomly sampled from $\mathcal{T}$ which ensures that our method is not trained using any additional data.

Our goal is also to minimize the expected value of the loss in Eq. (1) with respect to the model parameters $\theta$ via an algorithm such as stochastic gradient descent (SGD). Here, we go a step further and consider how the label imputation affects the ability of SGD to optimize the loss in Eq. (2).

We pose this as a meta-learning problem, which is derived next. We start by considering the loss value of the model $\Phi_\theta$ on the augmented training set $\mathcal{A}$:

$$\mathcal{C}^A = \sum_{\boldsymbol{x}\in\mathcal{T}} \ell(\Phi_\theta(\boldsymbol{x}), \boldsymbol{y}) + \sum_{\boldsymbol{x}^u\in\mathcal{U}} \ell(\Phi_\theta(\boldsymbol{x}^u), \boldsymbol{z}). \tag{3}$$

We first simulate a step of SGD using the loss in Eq. (3) to drive the update of the model parameters $\theta$. At step $t$, SGD updates the parameters for minimizing Eq. (3) as follows:

$$\hat{\theta}^{t+1} = \theta^t - \eta\nabla_\theta\left(\sum_{\boldsymbol{x}\in\mathcal{T}} \ell(\Phi_\theta(\boldsymbol{x}), \boldsymbol{y}) + \sum_{\boldsymbol{x}^u\in\mathcal{U}} \ell(\Phi_\theta(\boldsymbol{x}^u), \boldsymbol{z})\right), \tag{4}$$

where $\nabla_\theta$ is the gradient operator. We then wish to find $\boldsymbol{z}$ that minimizes the meta-validation objective $C^V(\hat{\theta}^{t+1})$ in Eq. (2) with $\hat{\theta}^{t+1}$. This corresponds to the following bilevel optimization problem

$$\min_{\boldsymbol{z}}\left(\sum_{\boldsymbol{x}\in\mathcal{V}} \ell(\Phi_{\hat{\theta}^{t+1}}(\boldsymbol{x}), \boldsymbol{y})\right), \text{ subject to } \hat{\theta}^{t+1} = \theta^t - \eta\nabla_\theta\left(\sum_{\boldsymbol{x}\in\mathcal{T}} \ell(\Phi_\theta(\boldsymbol{x}), \boldsymbol{y}) + \sum_{\boldsymbol{x}^u\in\mathcal{U}} \ell(\Phi_\theta(\boldsymbol{x}^u), \boldsymbol{z})\right). \tag{5}$$

Our goal is to learn predicted pseudo-labels that minimize the meta-validation loss in Eq. (5). To this end, we propose two options, denoted as **Option 1** and **Option 2**. In the former, we treat $\boldsymbol{z}$ as latent (or learnable) parameters and compute the gradients of the meta-validation loss w.r.t $\boldsymbol{z}$ to update $\boldsymbol{z}$, which is used to update model parameters $\theta$. In the latter, $\boldsymbol{z}$ is considered as the output of network $\Phi_\theta$ and the gradients are thus computed w.r.t the model parameters $\theta$ for updating $\theta$.

**Optimization.** The gradients of $C^V$ w.r.t $z$ for Option 1 and w.r.t $\theta$ for Option 2 are respectively be written as:

$$(\textbf{Option 1}) \qquad \frac{\partial C^V}{\partial z} = \sum_{x \in \mathcal{V}} \frac{\partial \ell(\Phi_{\hat{\theta}^{t+1}}(x), y)}{\partial \hat{\theta}^{t+1}} \cdot \left( \frac{\partial \theta^t}{\partial z} - \eta \nabla_z \nabla_\theta C^A \right) \qquad (6)$$

$$(\textbf{Option 2}) \qquad \frac{\partial C^V}{\partial \theta^t} = \frac{\partial C^V}{\partial z} \frac{\partial z}{\partial \theta^t} = \sum_{x \in \mathcal{V}} \frac{\partial \ell(\Phi_{\hat{\theta}^{t+1}}(x), y)}{\partial \hat{\theta}^{t+1}} \cdot \left( \mathcal{I} - \eta \nabla_\theta \nabla_\theta C^A \right) \qquad (7)$$

where $\mathcal{I}$ is identity matrix and $\nabla_\theta \nabla_\theta$ indicates a second order derivative which is supported by standard deep learning libraries such as Pytorch Paszke et al. (2017), Tensorflow Abadi et al. (2015).

We then use the gradients in Eq. (6) or Eq. (7) to update the model parameters such that the updated model can predict pseudo-labels that minimize the meta-validation loss. In Option 1, we update the pseudo-labels as $\hat{z} = z - \eta_z \frac{dC^V}{dz}$ and the updated pseudo-labels are then used to update the model parameters $\theta$ by minimizing the loss $\mathcal{C}^U = \sum_{x^u \in \mathcal{U}} \ell(\Phi_{\theta_t}(x_i^u), \hat{z}_i)$. In Option 2, the gradient in Eq. (7) is used to update the model parameters as $\theta^{t+1} = \theta^t - \eta \frac{\partial C^V}{\partial \theta^t}$

We further depict the optimization details of Option 1 and 2 in Alg. 1. We first estimate the $z$ with the current model $\theta^t$, update the model to optimize the loss $\mathcal{C}^A$ and then re-estimate $z$ with $\hat{\theta}^t$. This part is so far similar to the self-labeling method of Lee (2013). The loss function $\mathcal{C}^A$ can trivially be replaced by those used in recent work such as Mean-Teacher (Tarvainen & Valpola, 2017) or MixMatch (Berthelot et al., 2019); we applied our method in conjunction with all three loss functions and report results in Section 4. Next we apply the updated model to a mini-batch of unlabeled samples, compute the loss and simulate a SGD step w.r.t this loss. In Option 1, we initialize $z$ with the output of $\Phi_\theta(\tilde{x}^u)$ and update it by using the gradient from Eq. (6). In Option 2, the meta-update is computed w.r.t $\theta$ and used to update the model parameters by using the gradient from Eq. (7).

---

**Algorithm 1:** Pseudo-code of our method for two variants.

**Input**: $T, \mathcal{U}, \mathcal{V}$      ▷ `training set, unlabeled and meta-validation set resp.`
**Required**: model function $\Phi$ and its initialization parameters $\theta^0$, a batch of training data $\mathcal{B}_t^T = \{x_i, y_i\}_{i=1,\cdots,|\mathcal{B}_t^T|}$, a batch of unlabeled data $\mathcal{B}_t^\mathcal{U} = \{x_i^u\}_{i=1,\cdots,|\mathcal{B}_t^\mathcal{U}|}$ and another batch of training data (treated as meta-validation data during training) $\mathcal{B}_t^V = \{x_i, y_i\}_{i=1,\cdots,|\mathcal{B}_t^V|}$, learning rate $\alpha, \eta, \eta_z$, weight $\lambda$.

**for** $t = 0, \cdots, K-1$ **do**

  $z = \{z_i\}_{i=1,\cdots,|\mathcal{B}_t^\mathcal{U}|} = \{\Phi_{\theta^t}(\tilde{x}_i^u)\}_{i=1,\cdots,|\mathcal{B}_t^\mathcal{U}|}$     ▷ `estimate pseudo labels for`
  `unlabeled samples,` $\tilde{x}_i^u$ `is randomly perturbed input` $x_i^u$

  $\mathcal{C}^A = \sum_{i=1}^{|\mathcal{B}_t^T|} \ell(\Phi_{\theta^t}(x_i), y_i) + \lambda \sum_{i=1}^{|\mathcal{B}_t^\mathcal{U}|} \ell(\Phi_{\theta^t}(x_i^u), z_i)$

  $\hat{\theta}^t = \theta^t - \eta \frac{d\mathcal{C}^A}{d\theta^t}$     ▷ `update the model and move to the meta update`

  $z = \{z_i\}_{i=1,\cdots,|\mathcal{B}_t^\mathcal{U}|} = \{\Phi_{\hat{\theta}^t}(\tilde{x}_i^u)\}_{i=1,\cdots,|\mathcal{B}_t^\mathcal{U}|}$     ▷ `estimate pseudo labels using the`
  `updated model`

  $\mathcal{C} = \sum_{i=1}^{|\mathcal{B}_t^\mathcal{U}|} \ell(\Phi_{\hat{\theta}^t}(x_i^u), z_i)$

  $\hat{\theta}^{t+1} = \hat{\theta}^t - \alpha \frac{\partial \mathcal{C}}{\partial \hat{\theta}^t}$                     ▷ `simulate a SGD step`

  $\mathcal{C}^V(\hat{\theta}^{t+1}) = \sum_{i=1}^{|\mathcal{B}_t^V|} \ell(\Phi_{\hat{\theta}^{t+1}}(x_i), y_i)$     ▷ `evaluate on validation data`

  **if** *option 1* **then**

    $\hat{z} = z - \eta_z \frac{d\mathcal{C}^V}{dz}$                       ▷ `correct pseudo labels`

    $\mathcal{C}^U = \sum_{i=1}^{|\mathcal{B}_t^\mathcal{U}|} \ell(\Phi_{\theta_t}(x_i^u), \hat{z}_i)$

    $\theta^{t+1} = \hat{\theta}^t - \eta \frac{\partial \mathcal{C}^U}{\partial \hat{\theta}^t}$     ▷ `update the model with corrected pseudo labels`

  **else if** *option 2* **then**

    $\theta^{t+1} = \hat{\theta}^t - \eta \frac{\partial \mathcal{C}^V}{\partial \hat{\theta}^t}$     ▷ `update the model directly with meta-gradient`

  **Output**: $\theta^K$

**end**

---

## 4 EXPERIMENTS

We evaluate the performance of our method on multiple classification and regression benchmarks, and analyze the results below. Note that we use the validation sets only for tuning hyperparameters

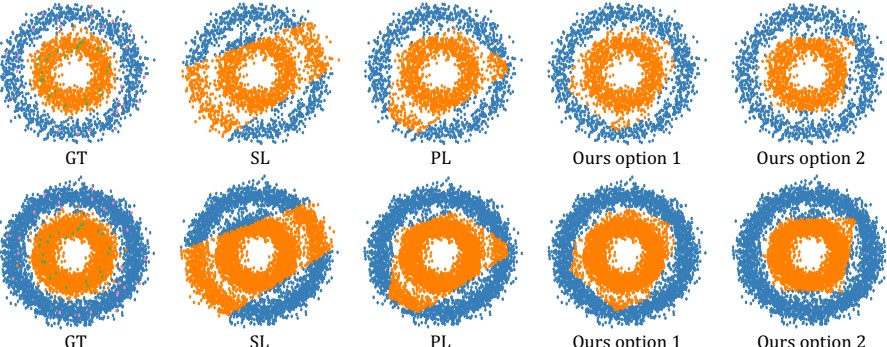

Figure 1: Results on the toy binary classification task. The top and bottom rows correspond to the predictions on the test and unlabeled data respectively. Points in green and pink are labeled training samples. Best seen in color.

such as learning rates and for early stopping. The meta-validation sets ($C^V$ in Eq. (5)) are sampled from the training set in all our experiments as in Rajeswaran et al. (2019); Finn et al. (2017); Vinyals et al. (2016); Sung et al. (2018). During training, we sample two mini-batches of data from the same training set at each iteration: one acts as training and the other one acts as the meta-validation set to ensure that our method is not trained on more data than the baselines. We train the model on the former and optimize the pseudo labels on the latter. [1]

### 4.1 CLASSIFICATION RESULTS

**Toy datasets.** We first validate our method on a synthetic binary classification dataset, two noisy concentric circles[2], where each circle corresponds to a class. To this end, we generate 10,000 samples and randomly pick 50, 2500 and 1000 for training, testing and validation sets respectively and use the rest of the samples as unlabeled data. Figure 1 illustrates the dataset where the labels for both test (top row) and unlabeled data (bottom row) are indicated in blue and orange colors. The labeled training samples for two classes are shown in green and pink in GT.

**Experiment 1.** In Fig. 1 we depict the predictions of two baselines: supervised learning (SL) which is trained only on the labeled training set (*i.e.* green and pink samples), Pseudo-Labeling (Lee, 2013) (PL) which iteratively first trains a network on the labeled and unlabeled data and then re-labels the unlabeled ones, and two variants of our method. For all the methods, we use a shallow small capacity network containing two fully-connected layers ($2 \xrightarrow{\text{fc}} 4 \xrightarrow{\text{fc}} 2$), one leaky ReLU activation layer in-between and a sigmoid function at the end. We first observe that SL misclassifies most of the outer circle points and predicts them as inner circle labels. This is due to the limited labeled data and its non-uniform spread over the input space. We see that PL significantly improves over SL by leveraging the unlabeled data. However many of its predictions are still inaccurate, especially in the regions of low label density and high ambiguity. This occurs because the iterative relabeling procedure is sensitive to the initial labeling of the unlabeled samples and thus it gets stuck in a non-optimal minimum. Both the variants of our method largely overcome the bias in the training data by labeling the unlabeled samples in a way that they lead to accurate predictions on the meta-validation set. This shows that the meta-updates are successful to correct the imputed labels based on the signal from the meta-validation set (see Fig. 1 bottom row) and to prevent our method to simply overfit into the pseudo-labels. While both variants agree on the most of the samples, they slightly differ in the low-density regions (*i.e.* the area between the circles).

**CIFAR-10 & -100.** We also evaluate our method on the CIFAR image classification benchmarks (Krizhevsky et al., 2009) that are commonly used for evaluation of both supervised and semi-supervised classification. Both datasets contain 50,000 and 10,000 training and testing samples respectively. In our experiments, we strictly follow the training and testing protocol for semi-

---

[1]Our implementation in PyTorch will be available at `https://anonymous.4open.science/r/a4721095-8266-4038-9cc6-8791ef61c610/`.

[2]https://scikit-learn.org

supervised learning which is proposed by the previous work (Oliver et al., 2018; Berthelot et al., 2019) where 5000 of training samples are used as validation data, the remaining 45,000 training samples are split into labeled and unlabeled training sets. As in Oliver et al. (2018), we randomly pick $|\mathcal{T}|$ samples as labeled data and the rest ($|\mathcal{U}| = 45,000 - |\mathcal{T}|$ samples) are unlabeled data. We report results for multiple training data-regimes $|\mathcal{T}| = 250, 500, 1000, 2000, 4000$ on CIFAR-10 and $|\mathcal{T}| = 1000, 2000, 3000, 4000, 5000$ on CIFAR-100. Note that we do *not* use the validation data in training of the model parameters but only for hyperparameter selection.

**Experiment 2.** First we compare our method to the SL and PL baselines for various number of training samples and report their performance in Table 1. We use a 13-layer conv-net as the classifier for all the methods (see Appendix A for details). First we observe that PL improves over SL by leveraging the unlabeled data in all the settings. Both our methods (option-1/2) that builds on PL achieves substantial performance gains over PL. Interestingly the relative improvements are higher in the more challenging case of when the labeled data is limited. We also observe that option-2, which involves updating the model parameters with meta-gradient, is a better strategy for these benchmarks.

| | CIFAR-10 | | | | | CIFAR-100 | | | | |
|---|---|---|---|---|---|---|---|---|---|---|
| #labels | 250 | 500 | 1000 | 2000 | 4000 | 1000 | 2000 | 3000 | 4000 | 5000 |
| SL | 57.95 | 46.06 | 37.21 | 25.32 | 19.03 | 82.32 | 72.08 | 65.06 | 61.81 | 57.90 |
| PL | 45.19 | 36.58 | 27.19 | 22.12 | 17.5 | 76.30 | 67.32 | 60.76 | 56.53 | 53.82 |
| PL+option1 | 43.74 | 35.76 | 26.46 | 21.14 | 16.81 | 75.67 | 66.03 | 60.36 | 56.28 | 53.01 |
| PL+option2 | **42.51** | **35.43** | **26.38** | **20.74** | **16.65** | **75.12** | **64.37** | **59.03** | **56.00** | **52.79** |

Table 1: Error percentage of testing set on CIFAR-10 and CIFAR-100.

**Experiment 3.** Here we show that our method can be incorporated to the state-of-the art methods such as Mean Teacher (MT) (Tarvainen & Valpola, 2017) and MixMatch (MM) (Berthelot et al., 2019) that use more sophisticated backbone networks, augmentation and regularization strategies, and also boost their performance. For this experiment, we follow the implementation of Berthelot et al. (2019) – we adopt a more competitive backbone, WideResNet-28-2, use the Adam optimizer along with standard data augmentation techniques (see Appendix A for more details).

| | CIFAR10 | | | | | CIFAR100 | | | | |
|---|---|---|---|---|---|---|---|---|---|---|
| #labels | 250 | 500 | 1000 | 2000 | 4000 | 1000 | 2000 | 3000 | 4000 | 5000 |
| $\Pi$ model | 53.02 | 41.82 | 31.53 | 23.07 | 17.41 | - - | - - | - - | - - | - - |
| PseudoLabel | 49.98 | 40.55 | 30.91 | 21.96 | 16.21 | - - | - - | - - | - - | - - |
| MixUp | 47.43 | 36.17 | 25.72 | 18.14 | 13.15 | - - | - - | - - | - - | - - |
| VAT | 36.03 | 26.11 | 18.68 | 14.40 | 11.05 | - - | - - | - - | - - | - - |
| MT | 38.90 | 28.45 | 19.36 | 12.40 | **10.02** | 76.78 | 64.01 | 55.30 | 52.03 | 48.28 |
| MT + option1 | 34.72 | 27.25 | 18.07 | 12.70 | 10.36 | 75.84 | 62.88 | 57.22 | 51.75 | 48.49 |
| MT + option2 | **27.72** | **21.05** | **14.93** | **11.91** | 10.31 | **75.24** | **61.45** | **54.56** | **50.31** | **45.40** |
| MM | 12.33 | **10.14** | **8.68** | **7.79** | 6.70 | 65.45 | 51.14 | **44.96** | 41.51 | **38.16** |
| MM + option1 | 11.60 | 10.24 | 9.60 | 8.37 | **6.59** | 63.62 | **50.53** | 45.67 | **40.76** | 38.88 |
| MM + option2 | **10.92** | 10.80 | 9.54 | 8.37 | 7.06 | **61.98** | 51.73 | 46.45 | 41.25 | 39.35 |

Table 2: Error percentage of testing set on CIFAR-10 and CIFAR-100 and comparison to the state-of-the-art semi-supervised learning methods.

Table 2 depicts the classification error rate for the several state-of-the-art techniques including $\Pi$ model (Laine & Aila, 2017), PL (Lee, 2013), VAT (Miyato et al., 2018), MT (Tarvainen & Valpola, 2017) and MM (Berthelot et al., 2019). Note that some methods do not report results on CIFAR-100. All the results except MT and MM are taken from the original papers. As our methods are built on MM and MT, we show the results of our own implementation which are on par with the published ones. From the table, we see that our method achieves significant improvement over MT baseline, especially in the low label regime, up-to 11 points in case of 250 labels in CIFAR-10. Again our second variant consistently outperforms the first one when used with the MT in both CIFAR-10 and -100, whereas the more competitive MM baseline already produces accurate pseudo-labels in CIFAR-10 and so the two options perform comparably. The reason is, in option2, we obtain the

meta gradient on the model parameters and update the model directly, whereas in option1, we firstly update the pseudo labels and then train the model on them. Though updating the pseudo labels can improve the performance (e.g. MT + option1 vs MT), some of the labels can still be noisy after the update and optimizing the model on the noisy ones may degrade the performance. In contrast, in option2, the meta gradient is applied to the network parameters directly, which alleviates the potential pseudo-label noise.

In case of MM, our method is able to boost its performance only in case of few labels (250 for CIFAR-10 and 1000 for CIFAR-100) and does comparable and slightly worse in case of more labels. We believe that it is harder to improve the performance of MM in the CIFAR, as its performance approaches to the supervised counterpart quickly. We leave the evaluation of MM on a more challenging benchmark as future work and below we show results for tasks where MM is not applicable.

## 4.2 REGRESSION RESULTS

**AFLW.** Next we move to a regression task and use the Annotated Facial Landmarks in the Wild (AFLW) dataset (Koestinger et al., 2011; Zhang et al., 2015) where we aim at predicting 5 landmarks' location of faces in images. The AFLW is originally designed for supervised facial landmark detection. We use the official train and test splits, randomly pick 10% of samples of the original training set as the validation set *only* for hyperparameter tuning and early stopping, and use the rest of this data as labeled and unlabeled data in our experiments. We evaluate the baselines and our method for 1%, 2%, 5%, 10% of training data as labeled data and report the standard Mean Square Error (MSE) normalized by the inter-ocular distance as in Zhang et al. (2015).

**Experiment 4.** For the regression task, we adopt the TCDCN backbone architecture in Zhang et al. (2015) and SGD optimizer, use standard data augmentation. We train all methods for 22500 steps. and set the initial learning rate to 0.03 and reduce it by 0.1 every 750 steps. The momentum and the weight decay are set to 0.9 and 0.0005. Here we use SL, PL and MT as the baselines and also build our method on both the PL and MT. Note that MM is not applicable to this task, as mixing up two face images doubles the number of landmarks. Table 3 depicts the results for the baselines and ours in terms of mean error rate.

| #labels | 1% | 2% | 5% | 10% | 100% |
|---|---|---|---|---|---|
| SL | 16.31 | 14.32 | 11.78 | 10.30 | 7.83 |
| PL | 12.97 | 11.65 | 9.70 | 9.01 | - - |
| Ours: PL + option1 | **11.86** | **10.86** | **9.46** | 8.93 | - - |
| Ours: PL + option2 | 12.77 | 11.47 | 9.85 | **8.90** | - - |
| MT | 12.87 | 11.38 | 9.85 | 8.93 | - - |
| Ours: MT + option1 | **11.85** | **10.74** | **9.43** | **8.58** | - - |
| Ours: MT + option2 | 12.45 | 11.27 | 9.69 | 8.79 | - - |

Table 3: Mean error percentage of testing set on AFLW for the facial landmark detection.

First we observe that the supervised learning on 1% of the labels is very challenging and obtains only 16.31% which is on par with the performance of simply taking the mean of each facial landmark over all training samples (16.58%). As expected, using the unlabeled face images is beneficial and both PL as well as MT methods significantly improves over SL. Our method achieves consistent improvement over PL and MT for different portions of labels. This strongly suggests that our method is able to refine the estimated landmarks of PL and MT on the unlabeled images and further improve its performance. We also analyze the effect of meta-updates during training our model in Fig. 2. To this end, we first plot the regression loss on the meta-validation batch before and after the meta-update in the left figure. This clearly shows that updating the pseudo landmark positions in the unlabeled images lead to a better accuracy on the meta-validation samples. Second we show the test loss for the same models in the right plot. It is clear that the meta-updated model does not overfit to the meta-validation set and generalizes better to the test images. We also visualize the effect of meta-updates on the landmarks on the example test images and observe that the meta-updates help them to get closer to the ground-truth ones.

Though both option-1 and 2 improve over the baselines, here option-1 outperforms option-2 on the regression problem which is in contrast to the classification tasks above. This is possibly due to the fact that the output space of the landmarks is continuous and less constrained than the label space for classification which makes the option-2 more prone to overfit to the validation set. A promising direction which is worth investigating in future is to alleviate the overfitting by using a regularizer to enforce the structure in the output space.

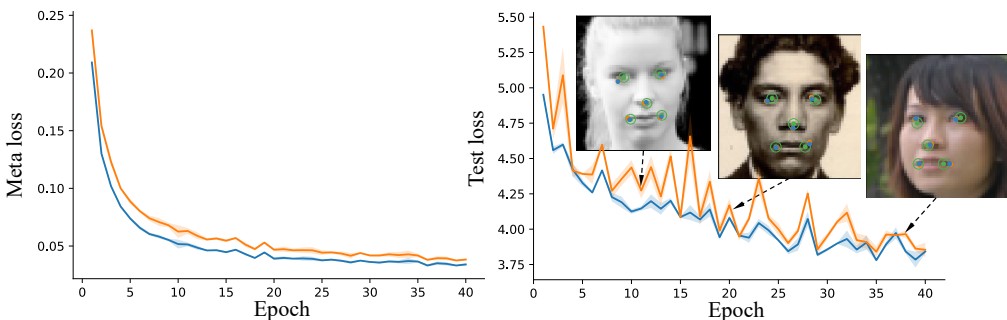

Figure 2: Illustration of the meta update's effect on facial landmarks detection. Cyan curves are the loss of the model after the meta update while orange curves are the model's loss before the meta update. Best seen in color.

We also illustrate success/failure cases on the test images in Fig. 3 and depict the ground-truth, predicted landmarks of MT and our method when trained with 1% of the labeled data. The performance difference is visually significant and our method outputs more accurate landmarks than MT. The bottom row shows the cases of extreme pose variation and occlusion where both MT and ours fail to achieve accurate predictions.

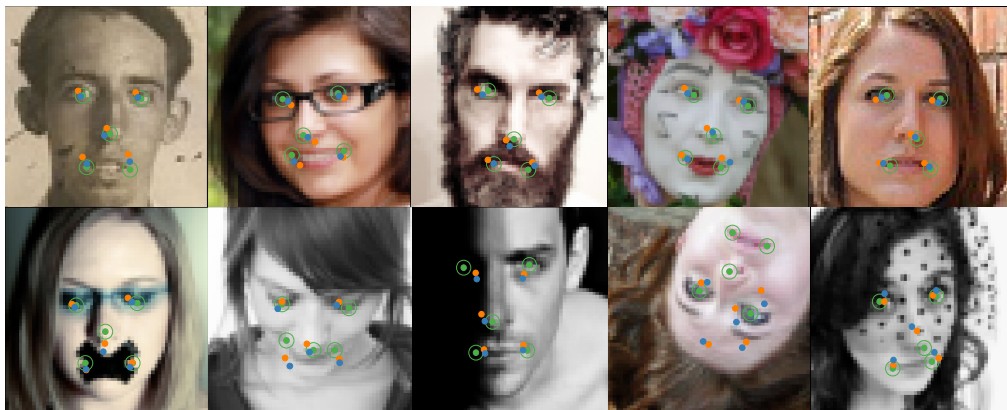

Figure 3: Success (top row) and failure cases (bottom row) in AFLW for facial landmark detection. Green points with circles, cyan points and orange points are ground-truth landmarks, predictions of our method (option-1) and Mean Teacher. Best seen in color.

## 5 CONCLUSION

In this paper we have propose a general semi-supervised learning framework, it learns to impute the labels of the unlabeled data such that the training a deep network on these labels improves its generalization ability. Our method can easily be used in conjunction with several state-of-the-art semi-supervised methods and extended to multiple classification and regression tasks such as image classification and facial landmark detection. We show that our method achieves significant performance gains over competitive baselines in challenging benchmarks, especially when the labeled data is scarce. As future work, we plan to extend our method to semi-supervised learning in structured output problems.

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

# A APPENDIX

To evaluate our method with existing semi-supervised learning, we implement our method and other methods in Pytorch.

## A.1 TOY EXPERIMENTS

**Experiment 1.** To conduct experiments on the concentric circles dataset, we use a small network containing two fully-connected layers ($2 \xrightarrow{\text{fc}} 4 \xrightarrow{\text{fc}} 2$) and one leaky relu activation layer. We compare our option 1 and option 2 to the Pseudo Label (PL) Lee (2013) and supervised learning approach and we train all methods for $20 \times 20$ steps (*i.e.* 20 epochs and 20 steps per epoch). We use Adam as the optimizer and the learning rate is 0.03. The maximum of the unsupervised loss weight $\lambda$ for both PL and Ours is set to 1 and we adopt the linear-schedule for increasing the weight on the unsupervised loss. Specifically, the $\lambda$ is initialized at 0 and increases to 1 gradually in $20 \times 5$ steps by linear-schedule as Berthelot et al. (2019). During the meta update in both our option1 and option2, we estimate the pseudo labels of unlabeled samples by gumbel softmax and compute the Mean Square Error (MSE) between prediction and pseudo labels as the unlabeled loss. $\eta_z$ is 1 for all experiments.

## A.2 CLASSIFICATION EXPERIMENTS ON CIFAR-10 & -100

For all experiments on both CIFAR-10 and CIFAR-100, we follow the optimizing strategy in Berthelot et al. (2019), i.e. we adopt Adam as the optimizer and fix the learning rate. We evaluate the model using an exponential moving average of the learned models' parameters with a decay rate of 0.999. In addition, the weight decay is set to 0.02 for all methods. We report the median error of the last 20 epochs for comparisons. For image preprocessing, we use standard data augmentation such as standard normalization and flipping, random crop as Berthelot et al. (2019). During the meta update in both our option1 and option2, we predict the pseudo labels of unlabeled images by applying softmax on the augmentation of the original unlabeled image and compute the Mean Square Error (MSE) between prediction on the original image and pseudo labels as the unlabeled loss as in Berthelot et al. (2019).

**Experiment 2.** We use a 13-layer conv-net Tarvainen & Valpola (2017); Laine & Aila (2017) in the experiments where we compare our method to the SL and PL on both CIFAR-10 and CIFAR-100. The architecture of the network is illustrated in Fig. 4. In addition to this, we apply batch normalization on each convolutional and fully connected layers. We use Leaky Relu with negative slope ($\alpha = 0.1$) as the nonlinear activation function on each convolutional layers. On CIFAR-10, the batch size and learning rate is set to 50 and 0.003, respectively, as in Tarvainen & Valpola (2017) while on CIFAR-100, the batch size is 128. We train each method for $40 \times 1000$ steps. For unsupervised loss weight $\lambda$, we use the linear-schedule and it increases gradually to 75 in $40 \times (1000 \times 0.4)$ steps (for both the PL and Ours).

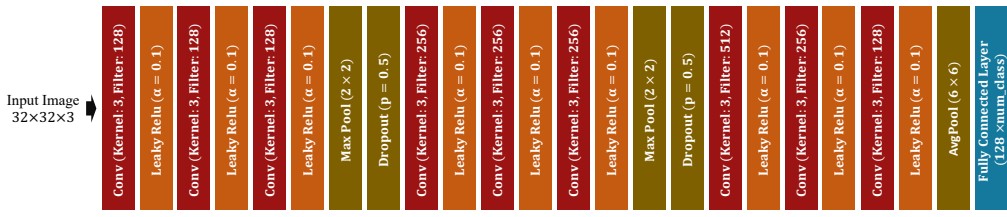

Figure 4: Illustration of the 13-layer Convolutional Network.

**Effect of Meta-validation Batch Size.** To study the influence of the meta-validation batch size, we set the batch size of the meta-validation mini-batch to 25, 50, 100 (Note, in experiment 2 on CIFAR-10, the batch size of the labeled, unlabeled and meta-validation data are set to 50 as in Tarvainen & Valpola (2017); Laine & Aila (2017)) and report the results of two variants of our method applied to PL for 250 labels case on CIFAR-10 in Fig. 5. More specifically, our option1 using 25, 50, 100 samples as meta-validation data at each training iteration obtain 43.4 %, 43.74 % and 43.72 %, respectively while option2 attains 42.7 %, 42.51 % and 42.47 %. These results

again strongly verify that both option1 and option2 achieve consistent improvements over the PL. In addition, it is clear that the performance of our method is not sensitive to the batch size of the meta-validation mini-batch (*i.e.* less than 0.4 %).

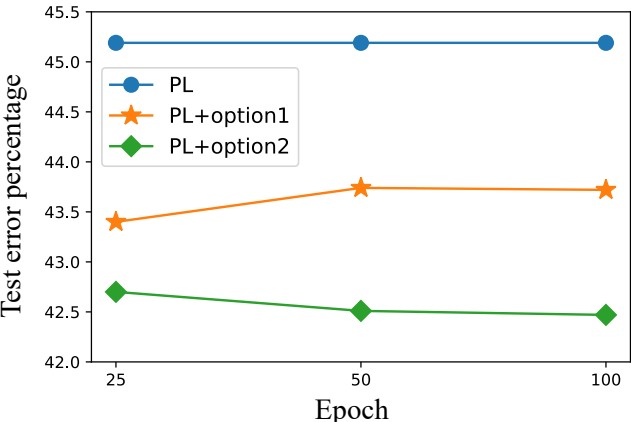

Figure 5: Evaluation of the meta-validation mini-batch's size (250 labels on CIFAR-10).

**Experiment 3.** We adopt the WideResNet-28-2 as the network for all methods as Berthelot et al. (2019). The batch size on CIFAR-10 is set to 32 and it is 128 on CIFAR-100. The learning rate is initialized as 0.002 and fixed. We use the same consistency weight and ramp-up schedule (linear-schedule) to increase the unsupervised loss weight $\lambda$ as Berthelot et al. (2019).

### A.3 REGRESSION EXPERIMENTS ON AFLW

**Experiment 4.** We adopt the TCDCN proposed in Zhang et al. (2015) as the network and SGD as optimizer. An illustration of the TCDCN's architecture is shown in Fig. 6. We adapt the Pseudo Label (PL), Mean Teacher (MT) and the supervised learning methods as the baselines and our method is built on PL as well as MT. To estimate the loss on an unlabeled image, we firstly crop an image from the original image by moving the cropping window a random number of pixels. We then estimate the location of landmarks on the augmented image and subtract the number of moving pixels, resulting in the pseudo label for the original image. We then apply MSE to the prediction of the original image and the pseudo labels to estimate the loss. We use the linear-schedule to update the unsupervised loss weight $\lambda$ to 1 in 9000 steps.

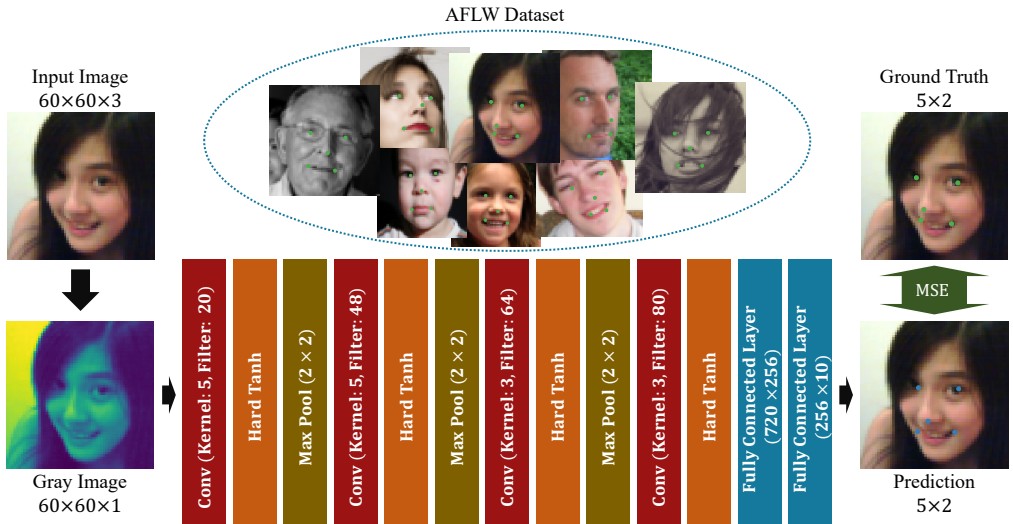

Figure 6: Illustration of the TCDCN.

