# OpenReview forum: "LEARNING TO IMPUTE: A GENERAL FRAMEWORK FOR SEMI-SUPERVISED LEARNING"
_ICLR.cc/2020/Conference — Reject_

### Official Review · AnonReviewer3 · 2019-10-21
**Official Blind Review #3**

**Rating:** 3

**Review:**

This paper proposes a semi-supervised approach to impute the labels of unlabeled samples such that a network achieves better generalization when it is trained on these labels. The proposed strategy can be easily used to improve the state-of-the-art semi-supervised methods. It mainly uses a validation data set to evaluate the updating rules of the unlabeled samples with pseudo-labels. The proposed method is applicable to both classification and regression problems including image classification and facial landmark detection tasks, which has shown in the experiments. But the following should be improved in the following aspects:
[1] In the proposed method, the model parameters are updated both on the unlabeled samples and validation data set. The experimental results show that such a strategy is effective to improve the performance of the state-of-the-art method. But why the strategy is effective should be further analyzed.
[2] How the validation data can improve the generalization ability of the model should be given with theoretical analysis. Whether the size of the validation data has a great influence?
[3] Some experimental settings are not clear. In the experiments, how many unlabeled data is labeled with pseudo-labels. For different size of the unlabeled data, how many samples should be used in the validation data to evaluate the model with pseudo labeled samples.
[4] How to divide the training data and the validation data? Whether the validation data need much more that the training data? How about the results only with all the labeled samples, which can further improve the confidence of the proposed method.


**Experience Assessment:**

I have published one or two papers in this area.

**Review Assessment: Checking Correctness Of Derivations And Theory:**

N/A

**Review Assessment: Checking Correctness Of Experiments:**

I did not assess the experiments.

**Review Assessment: Thoroughness In Paper Reading:**

I read the paper at least twice and used my best judgement in assessing the paper.

---

> ### Author Response · Authors · 2019-11-15
> **Response to Reviewer 3**
>
> We thank the reviewer for the feedback and respond to the individual points below.
>
> Q1: why the strategy is effective should be further analyzed.
> RE: Our hypothesis is that models trained on more accurate labels will yield higher performance for a given task. Assuming that evaluation on a meta-validation set is a good proxy of generalization performance, pseudo-labels that improve the model’s accuracy on the meta-validation set should be close to the ground-truth labels and thus training on the pseudo-labels should improve the generalization performance of the model. We support this hypothesis in the analysis at Figure 2. We show that the updated pseudo-labels obtain lower loss on both the meta-validation set and test set.
>
> Q2: How the validation data can improve the generalization ability of the model should be given with theoretical analysis. Whether the size of the validation data has a great influence?
> RE: We agree that a theoretical analysis on the generalization ability of our method would be interesting, but it is beyond the scope of this paper. In this work, the validation data are used only for early stopping and hyper-parameter tuning. Similar to previous meta-learning methods (Rajeswaran et al. 2019; Finn et al. 2017; Vinyals et al. 2016; Sung et al. 2018), during training, we sample two mini-batches of data from the same training set at each iteration: one acts as training and the other one acts as the meta-validation set to ensure that our method is not trained on more data than the baselines. We train the model on the former and optimize the pseudo labels on the latter. Thus the effective size of the validation set is the half of the training set at each epoch. This is now clearly indicated in the text (see first paragraph of Experiments).
>
> We also report an additional experiment to study the influence of meta-validation batch size in Appendix A.2. “To study the effect of the meta-validation batch size, we set the batch size of the meta-validation mini-batch to 25, 50 and 100 (Note, in experiment 2 on CIFAR-10, the batch size of the labeled, unlabeled and meta-validation data are set to 50 as in Tarvainen & Valpola (2017); Laine & Aila (2017)) and report the results of two variants of our method applied to PL in the 250 labels case on CIFAR-10 in Fig. 5. More specifically, our option1 using 25, 50, 100 samples as meta-validation data at each training iteration obtain 43.4 %, 43.74 % and 43.72 %, respectively while option2 attains 42.7 %, 42.51 % and 42.47 %. These results again strongly verify that both option1 and option2 achieve consistent improvements over the PL. In addition, it is clear that the performance of our method is not sensitive to the batch size of the meta-validation mini-batch (i.e. less than 0.4 %).”
>
> Q3: Some experimental settings are not clear: how many unlabeled data, how many samples should be used in the validation data to evaluate the model with pseudo labeled samples?
> RE: We follow the protocol in (Oliver et al. 2018), as explained in detail in Section 4.1:  “We also evaluate our method on the CIFAR image classification benchmarks (Krizhevsky et al., 2009) that are commonly used for evaluation of both supervised and semi-supervised classification. Both datasets contain 50,000 and 10,000 training and testing samples respectively. In our experiments, we strictly follow the training and testing protocol for semi- supervised learning which is proposed by the previous work (Oliver et al., 2018; Berthelot et al., 2019) where 5000 of training samples are used as validation data, the remaining 45,000 training samples are split into labeled and unlabeled training sets. As in Oliver et al. (2018), we randomly pick |T| samples as labeled data and the rest (|U| = 45, 000 − |T| samples) are unlabeled data. We report results for multiple training data-regimes |T| = 250, 500, 1000, 2000, 4000 on CIFAR-10 and |T| = 1000, 2000, 3000, 4000, 5000 on CIFAR-100. Note that we do not use the validation data in training of the model parameters but only for hyperparameter selection.”
>
> Q4: How to divide the training data and the validation data? Whether the validation data need much more that the training data? How about the results only with all the labeled samples, which can further improve the confidence of the proposed method.
> RE: We follow the experimental setting (training and validation set) that is proposed by (Oliver et al. 2018). As stated above, the validation data in this work is used for early stopping and hyper-parameter tuning only. The training and meta-validation sets are effectively the same in our experiments. The errors for the fully supervised baselines that are trained on the whole training set are 4.17%, 25.44%, 7.83% for CIFAR-10, CIFAR-100 and AFLW respectively. These numbers set the upper bound on performance for the semi-supervised methods.

---

### Official Review · AnonReviewer4 · 2019-10-31
**Official Blind Review #4**

**Rating:** 3

**Review:**

This paper looks into problem of semi-supervised learning and in order to be mindful of generalization on the unlabeled data, they add a term to the loss function which includes loss on imputed labels.
I have 3 main concerns with the paper
1. The authors mention that  the meta-validation set is a random subset of train set. and they check the final performance on meta-validation set. This does not seem a right way to measure performance of the model as meta-validation set is already used in training. The set of labeled points should be partitioned into train and meta-validation set.

2. The derivation of the updates given the added term to the loss. In option 1, the authors mention they use Eqn. 8 to update z, while Eqn 8. has the reverse information.

3.In option 2, z = \sigmoid(\Phi_\theta) and reducing the loss on z, l( \sigmoid(\Phi_\theta) ,  \Phi_\theta), does not look very meaningful. trying to get  \Phi_\theta close to its sigmoid means getting it close to zero. but we do not know what is the label for unlabeled data, so why getting the label close to zero?
Also the authors mention that second order derivatives will come to play without any explanation. I suggest spending more effort on explaining the problem formulation as that's the core of the paper.

More comments:
* As mentioned above the problem formulation is not clean and there are unjustified choice there. Moreover, the experiment results are mostly declared without any justification (for example, the proposed method does not always lead to improvement and not all cases are explained. The authors only note that the method works well in low data regime).

* In the first experiment PL is compared to two cases of the proposed algorithm whereas in other experiments PL is compared to combining PL with versions of the proposed method. Is there a reason for this?

* The models used as baseline are only explained briefly in the last page of the paper, while being used multiple time in the experiment section. This is not good writing practice.

**Experience Assessment:**

I have read many papers in this area.

**Review Assessment: Checking Correctness Of Derivations And Theory:**

I assessed the sensibility of the derivations and theory.

**Review Assessment: Checking Correctness Of Experiments:**

I assessed the sensibility of the experiments.

**Review Assessment: Thoroughness In Paper Reading:**

I read the paper at least twice and used my best judgement in assessing the paper.

---

> ### Author Response · Authors · 2019-11-15
> **Response to Reviewer 4**
>
> We thank the reviewer for the feedback and respond to the individual points below.
>
> Q1: … check the final performance on meta-validation set … does not seem a right way to measure performance of the model as meta-validation set is already used in training. The set of labeled points should be partitioned into train and meta-validation set.
> RE: All our results in Table 1-3 are obtained on the test sets as per previous work. To clarify, the meta-validation set is indeed randomly sampled from the labeled training set at each epoch as in previous meta-learning methods (Rajeswaran et al. 2019; Finn et al. 2017; Vinyals et al. 2016; Sung et al. 2018). It is then used during training to compute the gradients with respect to the estimated pseudo-labels as indicated in Equations 6 and 7.
>
> Q2: The derivation of the updates given the added term to the loss. In option 1, the authors mention they use Eqn. 8 to update z, while Eqn 8. has the reverse information.
> RE: The previous Eqn. 8 provides the update \delta_z with respect to z, i.e.  z_{t+1} = z_{t} - \eta_z \delta_z. We have updated Section 3 to detail how we optimize the model with the second derivative in Equation 6 (Option 1) or the one in Equation 7 (Option 2). This is also detailed in Algorithm 1.
>
> Q3a: In option 2, z = \sigmoid(\Phi_\theta) and reducing the loss on z, l( \sigmoid(\Phi_\theta) ,  \Phi_\theta), does not look very meaningful. trying to get \Phi_\theta close to its sigmoid means getting it close to zero. but we do not know what is the label for unlabeled data, so why getting the label close to zero?
> RE: To clarify, the \sigmoid symbol is used to represent the softmax or Gumbel-softmax operator over the object categories and not the sigmoid function, as previously explained in Option 2 at page 3.
>
> In addition, we updated the manuscript (Sec. 3) to clarify this point. We wish for the output of the network to be close to the output of a perturbed model (e.g. via perturbed inputs or  perturbations like dropout); this is a form of classifier smoothing. Our implementation uses different random crops of input images as perturbations when our inputs are images. In the toy experiments, where the input samples are 2-d points, we used the Gumbel-softmax function to compute the pseudo-labels, as it outputs hard-labels (or one-hot probabilities) in a differentiable manner (Appendix A.1).
>
> Q3b: Also the authors mention that second order derivatives will come to play without any explanation. I suggest spending more effort on explaining the problem formulation as that's the core of the paper.
> RE: We now describe the formulation with the second derivative explicitly in Equations 6 and 7. We also detail how we optimize the model with the second derivative in Equation 6 (Option 1) or the one in Equation 7 (Option 2) in Sec. 3.
>
> Q4: As mentioned above the problem formulation is not clean and there are unjustified choice there. Moreover, the experiment results are mostly declared without any justification (for example, the proposed method does not always lead to improvement and not all cases are explained. The authors only note that the method works well in low data regime).
> RE: From the results, we see that our method does comparably (and sometimes better) when compared to baselines, when there are sufficient labeled samples; our method performs better in the low data regime, in which the learned model would be much noisier and the correction from the meta-validation has a potentially greater effect. We also believe that the most interesting comparisons are with very few labeled data points since it reveals the method’s sample efficiency which is central to SSL (Berthelot et al., 2019)
>
> Q5: In the first experiment PL is compared to two cases of the proposed algorithm whereas in other experiments PL is compared to combining PL with versions of the proposed method. Is there a reason for this?
> RE: In the first experiment, we mainly evaluate our method built on a vanilla semi-supervised technique, PL using a 13-layer conv-net and the results verify the efficacy on correcting the pseudo labels generated by PL. In the second experiment, we mainly incorporate our method to two state-of-the-art techniques Mean Teacher and MixMatch as they achieve much better performance than PL. The results in the second experiments on CIFAR-10&-100 also indicate that our method can successfully correct those noisy pseudo labels, especially in fewer label regime, leading to better performance. This also implies that our method can be incorporated into existing methods and is thus generic.
>
> Q6: The models used as baseline are only explained briefly in the last page of the paper, while being used multiple time in the experiment section. This is not good writing practice.
> RE: We have moved the related work section to Section 2 to address this.

---

### Official Review · AnonReviewer5 · 2019-10-31
**Official Blind Review #5**

**Rating:** 3

**Review:**

This paper uses a meta-learning approach to solve semi-supervised learning. The main idea is to simulate an SGD step on the loss of the meta-validation data and see how the model will perform if the pseudo-labels of unlabelled data are perturbed. Experiments on classification and regression problems show that the proposed method can improve over existing methods. The idea itself is intriguing but the derivation and some design choice are not very well-explained.

(1) The derivation from Eq.(3) to (4) is confusing. Note that in Eq.(3), the prediction \Phi_\theta also depends on \theta in addition to the pseudo-label z. When taking a step of SGD, the second term of Eq.(3) (with unlabelled data) will always be zero if both arguments of the loss (\Phi_\theta(x) and z_\theta(x)) change simultaneously. Eq.(4) somehow only considers the gradient of unsupervised loss, then the gradient would be zero because there is no incentive to deviate from the pseudo-label z. The pseudo-code does not help much. The update from \hat{\theta}^{t} to \hat{\theta}^{t+1} has the same issue: there is no incentive for \hat{\theta}^{t} to deviate because z is exactly produced by it.

(2) For classification problems, it is natural to use cross-entropy loss for the probability vector z. Are there any specific reasons for using Gumbel-softmax? In addition, using L2 loss for probability vectors (as mentioned in Appendix A) is known to be problematic as it may create exponentially many local minima (Auer et al, 1996).

(3) The recent work of Li et al. (2019) also considers iteratively improving pseudo-labels with meta-updates so it should be discussed and compared.

(4) Experiments
- What are the sizes of the meta-validation sets in the experiments?
- Error bars in the tables and Fig.2?
- The MM results in Table 2 are noticeably worse than the original results. For example, with 250 labeled data, MM achieved 11.08% in CIFAR-10 as reported in the original paper. (And 4000 labeled data can achieve 4.95%)
- It is said that option 2 is consistently better than option 1, which is not true for the MM baseline.
- 22500 training steps for Experiment 4 seems arbitrary. What are the candidates for the hyper-parameters?

Typos:
- In the first paragraph of Sec.2, one of the x and one of the y should be bold.
- Above Eq.(4), x^{U\in U} should be x^i \in U
- The transpose in Eq.(7) is not necessary
- It is said on page 6 that Fig.2 reports classification loss but the task is a regression problem.

Ref
- Auer, P., Herbster, M. and Warmuth, M.K., 1996. Exponentially many local minima for single neurons. In Advances in neural information processing systems (pp. 316-322).
- Li, X., Sun, Q., Liu, Y., Zheng, S., Chua, T.S. and Schiele, B., 2019. Learning to Self-Train for Semi-Supervised Few-Shot Classification. In Advances in neural information processing systems.

**Experience Assessment:**

I have read many papers in this area.

**Review Assessment: Checking Correctness Of Derivations And Theory:**

I assessed the sensibility of the derivations and theory.

**Review Assessment: Checking Correctness Of Experiments:**

I assessed the sensibility of the experiments.

**Review Assessment: Thoroughness In Paper Reading:**

I read the paper at least twice and used my best judgement in assessing the paper.

---

> ### Author Response · Authors · 2019-11-15
> **Response to Reviewer 5**
>
> We thank the reviewer for the feedback and respond to the individual points below.
>
> Q1. The derivation from Eq.(3) to (4) is confusing. ... the second term of Eq.(3) (with unlabelled data) will always be zero … no incentive to deviate from the pseudo-label z.
> RE: We have updated the manuscript (Sec. 3) to clarify this point. Setting the pseudo-labels to the output of the network (\ie z=\Phi_{\theta}(x^u)) yields zero loss and indeed no gradient for \ell(\Phi_\theta(x^u), z). Non-zero losses can be obtained by injecting stochasticity to the pseudo-label prediction (also a form of classifier smoothing) which can be realized in several ways: using drop-out layers in the network, adding random perturbations to the input (x^u) and/or output of the network (\Phi_\theta). Our implementation uses different random crops of input images to compute the output and pseudo-labels when our inputs are images.
>
> Q2. … any specific reasons for using Gumbel-softmax? … using L2 loss for probability vectors ... may create exponentially many local minima
> RE: We use the cross-entropy loss function for labeled data, whereas we adopt L2 loss function to penalize the inconsistency between the network's prediction (i.e. categories probability obtained by applying softmax operator to the network’s logits prediction) and the corresponding pseudo-label for unlabeled data. Note that L2 loss function is commonly used in the previous semi-supervised work (e.g. Tarvainen and Valpola, 2017; Berthelot et al. 2019) to measure such inconsistency. As studied by Berthelot et al. (2019) (see Section 3.4), the L2 loss is bounded and less sensitive to noisy labels compared to KL divergence.
>
> The Gumbel-softmax function is only used in the toy experiments to generate hard pseudo-labels such that the loss between predictions and pseudo-label is non-zero. Note that the Gumbel-softmax is differentiable and thus allows for back-propagation from the validation loss to the network parameters.
>
> Q3. The recent work of Li et al. (2019) should be discussed and compared
> RE: We have included a discussion of Li et al. (2019) in our related work section. This work is differs to ours in several ways. First, the problem setting is different: Li et al. (2019) focus on semi-supervised few-shot learning, where training, validation and test sets have separate label spaces and the goal is to learn a good model parameter initialization for efficiently adapting the network to new tasks with few labels. Second, they propose a meta-learning technique to initialize a self-training model and to filter out noisy pseudo labels; by contrast, our method does not require any such filtering process, uses all the pseudo labels and more importantly, learns to predict accurate pseudo-labels that achieve optimal performance on a meta-validation set.
>
> Q4a: What are the sizes of the meta-validation sets in the experiments?
> RE: As is standard practice in the field (Finn et al. 2017; Vinyals et al. 2016), during each training iteration we sample two same-sized (e.g. 64 samples for CIFAR-10) mini-batches from the training set, one for training and one as the meta-validation set. Thus, the effective size of the meta-validation set is the size of the training set.
>
> We have also added an additional experiment to study the influence of meta-validation batch size in Appendix A.2 (Please see Appendix for details).
>
> Q4b: Error bars in the tables and Fig.2?
> RE: As requested, we have added the standard deviation in Fig 2. As it is not feasible to run all the experiments in Table 1-3 multiple times in the short rebuttal period, these results will be included in the final version of the paper.
>
> Q4c: The MM results in Table 2 are noticeably worse than the original results.
> RE: The MM paper reports results for two backbone architectures, the standard WideResNet-28-2 (1.5M parameters) and a larger version of WideResNet (26M parameters). In this work, we adopt the standard WideResNet-28-2 for computational efficiency and obtained similar results to those reported in the original paper (6.70% vs 6.24% for 4000 labels).
>
> Q4d: option 2 is consistently better than option 1, which is not true for the MM baseline.
> RE: We have corrected this statement in the text as follows: “Again our second variant consistently outperforms the first one when used with MT in both CIFAR-10 and -100, whereas the more competitive MM baseline already produces accurate pseudo-labels in CIFAR-10 and so the two options perform comparably.”
>
> Q4e: 22500 training steps seems arbitrary.
> RE: The number of training steps are set such that the fully supervised baseline obtains similar performance with the one in (Zhang et al, 2015). This setting is used for all the methods in the regression experiments.
>
> Q5: Typos.
> RE: Corrected.

---

### Author Response · Authors · 2019-11-15
**Response to area chair and all reviewers**

We thank all reviewers for their valuable feedback.

The main concerns were clarification of:
1) how the meta-validation set was constructed (Reviewer#3, Reviewer#4, Reviewer#5)
2) the model formulation, in particular, how the prediction for pseudo-labels are designed and why they are different from the network’s output (Reviewer#4 and Reviewer#5).

We address these concerns below, and have also incorporated the requested clarifications and additional experiments into the manuscript. In summary we made the following changes:
1) we updated the manuscript to clarify the prediction for pseudo-labels and more clearly explain the model design choices
2) details of the meta-validation set are clearly indicated in Sec. 3 and at the beginning of the experiment section
3) we explicitly described the gradient updates the formulation with the second derivative in Equation 6 and 7 in Sec. 3 and detailed how we use the second derivative (meta gradient) for updating model
4) we investigate the effect of the meta-validation mini-batch size in Appendix A.2.

We note that reviewers did not raise any concerns about the originality and soundness of the proposed method. It is also pointed out by the reviewers that, our proposed semi-supervised method is applicable to both classification and regression problems and achieves improvements over respective state-of-the-art methods. We also would like to reiterate the contributions of our paper. We propose a new learning-to-learn method for semi-supervised learning. Our proposed meta learning method involves learning an update rule to label unlabeled training samples such that training our model using these predicted labels of unlabeled samples to improve its performance not only itself but also on a meta-validation set. In addition, our method is highly generic and can be easily incorporated to the state-of-the-art methods and boost their performance, in particular in fewer labels regime. Beyond this, we demonstrate that our method is applicable to both classification and regression problems including image classification and facial landmark detection tasks and achieves significant performance gains.

---

### Decision · Program_Chairs · 2019-12-19

**Decision:**

Reject

**Comment:**

There is insufficient support to recommend accepting this paper.  The reviewers unanimously criticize the quality of the exposition, noting that many key elements in the main development and experimental set up are not clear.  The significance of the contribution could be made stronger with some form of theoretical analysis.  The current paper lacks depth and insufficient justification for the proposed approach.  The submitted comments should be able to help the authors improve the paper.